# The Role of Micronutrients in Neurological Disorders

**DOI:** 10.3390/nu15194129

**Published:** 2023-09-25

**Authors:** Helena Lahoda Brodska, Jiri Klempir, Jan Zavora, Pavel Kohout

**Affiliations:** 1Institute of Medical Biochemistry and Laboratory Diagnostics, First Faculty of Medicine, Charles University and General University Hospital in Prague, U Nemocnice 499/2, 128 08 Prague, Czech Republic; helena.lahodabrodska@vfn.cz (H.L.B.); jan.zavora@vfn.cz (J.Z.); 2Department of Neurology and Centre of Clinical Neuroscience, First Faculty of Medicine, Charles University and General University Hospital in Prague, Katerinska 30, 120 00 Prague, Czech Republic; 3Department of Microbiology, Faculty of Medicine and Dentistry, Palacky University, Hnevotinska 3, 775 15 Olomouc, Czech Republic; 4Clinic of Internal Medicine, 3rd Faculty Medicine, Charles University and Thomayer University Hospital, Videnska 800, 140 59 Prague, Czech Republic; pavel.kohout@ftn.cz

**Keywords:** micronutrients, neurodegenerative disorders, autoimmune disorders, stroke, Parkinson’s disease, Huntington’s disease, neuropathy, multiple sclerosis, epilepsy

## Abstract

Trace elements and vitamins, collectively known as micronutrients, are essential for basic metabolic reactions in the human body. Their deficiency or, on the contrary, an increased amount can lead to serious disorders. Research in recent years has shown that long-term abnormal levels of micronutrients may be involved in the etiopathogenesis of some neurological diseases. Acute and chronic alterations in micronutrient levels may cause other serious complications in neurological diseases. Our aim was to summarize the knowledge about micronutrients in relation to selected neurological diseases and comment on their importance and the possibilities of therapeutic intervention in clinical practice.

## 1. Introduction

The aim of this review is to provide a comprehensive overview of the knowledge of the role of micronutrients in selected neurological diseases. This group of micronutrients consists of vitamins and trace elements. The quantity of micronutrients in the human body is very low (trace elements, up to 50 mg/kg; plasma levels of trace elements and vitamins are in the range of μmol/L to mg/L), but they are necessary for several essential functions of the organism [1]. The human body cannot synthesize vitamins at all or only in a small amount (vitamin D). Water-soluble vitamins (B, C) are not stored in the organism, unlike fat-soluble vitamins (A, D, E, K), which are stored in the organism, so a deficiency in fat-soluble vitamins may manifest later [2,3]. The loss of micronutrients or insufficient intake can lead to depletions that manifest in clinical and laboratory consequences. The main function of micronutrients is their catalytic effect in enzyme systems, either as cofactors or as components of metalloenzymes. Other essential roles of micronutrients are antioxidant activity, modulation of cellular immunity, and wound healing. Regarding the catalytic action, micronutrients affect the enzyme activities in intermedial metabolism (the use of amino acids, lipids, and carbohydrates) and regulate the genomic expression of structural proteins. Due to their antioxidant activity, micronutrients can neutralize free radical reactive oxygen species (ROS) and reactive nitrogen species (RNS). Stratified defense is influenced by passive scavengers of vitamins C and E, albumin, bilirubin, and uric acid, followed by enzymes contributing to radicals-to-H_2_O transformation, namely superoxide dismutase (SOD), glutathione peroxidase (GSHPx), and their cofactors Cu, Zn, Mn, and Se [4]. Micronutrients participate in the modulation of cellular immunity and tissue healing either directly (vitamin C, 1.25-OH vitamin D, Se, Zn) or by using substances created through their induction (cathelicidin LL-37).

## 2. Consequences in Neuropathology

The dysregulation of micronutrients leads to disorders of previously mentioned functions, such as damage to peripheral nerves (demyelination or axonal damage), damage to the central nervous system, and a special category of myeloneuropathy (damage to the peripheral and central nervous systems) [5]. The deficiency in individual micronutrients contributes to pathological changes in the development of the nervous system and can lead to the development of “nutritional” neuropathies. The potential influence of micronutrient imbalance and their subsequent supplementation on patient outcome has been less investigated in most neurodegenerative processes (Alzheimer’s disease, Parkinson’s disease, Huntington’s disease, and amyotrophic lateral sclerosis).

## 3. Micronutrients in Clinical Practice

In most cases, symptoms of micronutrient deficiency appear only after long-lasting depletion [1]. Symptoms are mostly nonspecific ranging from frequent infections, skin manifestation, microcytic (Fe, Cu) or macrocytic (B12) anemias, neuropathy, and anorexia, to characteristic symptoms of serious deficiencies like scurvy (vitamin C), osteomalacia (vitamin D), pellagra (niacin), hemorrhagic diseases (vitamin K), or night blindness (vitamin A). A long-lasting deficiency can manifest acutely as in the case of thiamine deficiency, after nutritional intervention (administration of energy substrate), or via lactic acidosis development.

Capillary leakage increases in pathological conditions within the system’s response to inflammation (infectious or noninfectious) due to the influence of released cytokines, which lead to micronutrients and their binding proteins transferring to different compartments. The plasma level of all micronutrients then decreases. Redistribution mechanisms are also involved in the decrease because the concentration of micronutrients with an antioxidant effect is highest at the site of inflammation. The increased requirement for micronutrients as intermediary metabolism cofactors and antioxidants contributes to the decrease. Their consumption increases with condition deterioration. It is important to acknowledge the potential increased loss of micronutrients with urine, drainages, and repeated hemodialysis, and in burn victims. The rapid drop in iron plasma levels is caused by complex mechanisms controlled by hepcidin production (see below).

The decrease in micronutrient plasma levels is rather a guess than a real detection. The levels of only a small number of micronutrients can be tested routinely. The micronutrient dilemma produces several questions.

### 3.1. Is it Necessary to Verify the Presumptive Decrease of Levels through Available Analysis? 

If it is possible, the answer to this question is “definitely” (vitamin D, active vitamin B12, folate, Fe, and, when indicated Se and Zn). It can be used to stratify patients in risk and monitor the success of the substitution.

### 3.2. If We Discover a Decrease in Levels, Is It a Real Deficiency or Purposeful Redistribution? 

It is essential to know the mechanism of the decrease. A rapid drop in Fe levels is the result of a sophisticated defensive reaction (see below). It is different in situations when the micronutrient decrease occurs as a result of an increased turnover (vitamin B1, C) or loss (Se, Zn, vitamin C).

### 3.3. Will Therapeutic Equalization of Plasma Levels Positively Affect Patient Prognosis? 

The need for supplementation stems from the mechanism of the decrease. The supplementation of Fe in acute conditions is counterproductive, unlike when supplementing vitamins B1 and C, Se, and others. In comparison with a healthy population, the levels of some micronutrients are significantly decreased in all neurodegenerative diseases, so substitution has its benefits (improvement in cognitive functions, prevention of diseases), and it is recommended as adjuvant therapy [6].

### 3.4. How Should the Potential Substitution Be Set Up? For Whom, What, How Much, and How? 

Substitution of micronutrients should be performed for patients with developing or existing risk of deficiency according to the current ESPEN (2022) guidelines [1]. Micronutrients should be administered every time macronutrients are administered. When administering sustenance via the enteral route, we must take into consideration the condition of the intestinal mucosa, particularly the actual quantity that can be absorbed, the limits of intestinal absorption (vitamin C) of individual micronutrients, and the competition principle (high dosage of Fe leads to zinc and copper resorption disorder, high supplementation of zinc complicates the absorption of copper since they have a common carrier). The situation of patients dependent on parenteral intake is mostly solved by adding a standard multicomponent mix of trace elements and vitamins that covers the recommended daily dose for most individuals. The fortification of selected micronutrients is possible through monotherapy (vitamins B1, C, and D and Se).

## 4. The Groups of Patients for Whom the Fortification of Selected Micronutrients Is Recommended

Fortification is recommended for ICU patients (increased usage, oxidative stress, redistribution), burn victims (oxidative stress, loss through burned area), patients with exudates, wound areas, and patients receiving repeated hemodialysis. Special attention is needed during substitution and the subsequent monitoring of the therapy success rate in patients with home parenteral nutrition (HPN). There is a special indication in neurology—acute and chronic conditions (see below).

## 5. Options for Micronutrient Administration

The enteral route is the most natural, either in the form of an ordinary meal, in the form of defined enteral nutrition using oral nutritional supplements (ONS), or defined enteral nutrition using a nasogastric/nasojejunal tube or gastrostomy. Enteral nourishment in recommended volume per day defined in this way includes not only a recommended daily dose of macronutrients but also of micronutrients. For the fortification of selected micronutrients, it is possible to take defined dietary supplements in a noncomplicated state. By administrating micronutrients this way, we need to take into consideration the principles of absorption of individual micronutrients (if we increase the quantity of one, we may preclude the absorption of a different one—the carrier competition—or we synergistically boost its absorption). For example, an increased intake of Fe reduces the resorption of Cu and Zn. Due to Zn and Cu having the same carrier during resorption from food, Cu is disadvantaged when the administration of Zn (dietary supplement) is increased, which leads to the depletion of Cu. This depletion manifests as microcytic anemia, which is resistant to the treatment of Fe and folic acid. This mechanism can be used therapeutically for Wilson’s disease by lowering the resorption of Cu with increased doses of Zn. Sometimes, synergy can be detected—administration of vitamin C endorses the absorption of Fe. It is also necessary to take into consideration the limited absorption ability of the GI tract (vitamin C). In patients with malabsorption and critically ill patients, it is necessary to consider that the whole quantity of micronutrients declared in the given amount of enteral nourishment is not absorbed fully. 

The second option is the administration through the parenteral route when using a range of pharmaceutical mixtures (see overview in Table 1 with clearly defined contents of micronutrients). Here, we use an administration within an all-in-one system, where it is necessary to adhere to certain limits (adding vitamins right before administration, keeping away from light, etc.). Then, there are preparations of individual vitamins (vitamin C, D, B12) intended to be administered either i.m. or i.v., which should be chosen during dose fortification.

## 6. Special Indication Categories in Neurology

The indication for micronutrient supplementation in neurology can be regarded from several perspectives. Firstly, the administration of micronutrients can be viewed as prevention against the development of their deficiency. The deficiency may then lead to the development of so-called nutritional neuropathy, or it can advance the symptoms of neurodegenerative illness development. The indication is based on the pathophysiology of the micronutrient effects, together with the etiopathogenesis of the illness. Primarily in inflammatory disease (infection but also systemic noninfective inflammatory response), depleted supplies should be replenished because of increased usage, to lower oxidative stress, etc. (delayed onset or, more precisely, development of dementias has been repeatedly reported in the preventive supplementation of vitamin D [3]). We often do not verify the levels of individual micronutrients: doses are administered per guideline daily amounts (GDAs), and higher pharmacological doses are mostly administered for early alignment of a deficit. We must pay attention to the supra-administration of vitamin C: it can limit intestinal absorption, and there is a risk of oxalate lithiasis and oxidative stress [1,7].

Secondly, another point of view is the effort to influence already-developed diseases with clinical symptoms through higher doses of the missing micronutrients in diseases with a familiar (Wernicke’s encephalopathy and the administration of thiamine) but not fully known pathophysiology of the effect (Alzheimer’s disease and administration of vitamin D). We either know that the symptoms are related to or directly caused by a fatal decrease in micronutrient(s), or we expect a positive effect on the progress of the certain disease from the pathophysiology of certain micronutrients.

The clarification of the possible clinical effects of micronutrient administration is usually very difficult and rarely so clear, like, for example, the rapid correction of lactic acidosis by the repeated administration of 500 mg of thiamine in the already mentioned Wernicke’s encephalopathy. The effect of vitamin B12 substitution in pernicious anemia has also been proven [1].

The positive effect of vitamin B group administration is more likely to be found in skin affections and less in neuritis where they are usually part of causal therapy. The administration of micronutrients in most indications is perceived as a necessary part of complex therapy, and its justification stems from the pathophysiology of individual micronutrients.

The quantification of the stored micronutrients using plasma level is not definite, especially with an ongoing inflammatory response, as stated above. We are not able to quantify the levels of a large number of micronutrients in routine work. Is this diagnosis of “circumstantial evidence” sufficient? Are we able to correctly interpret the evidence?

The recommendation on monitoring and supplementation of micronutrients in various neurological disorders is summarized in Table 2.

### 6.1. Alzheimer’s Disease 

Dementia in the elderly is most frequently caused by Alzheimer’s disease (AD). The disease process is manifested as neuritic plaques, extracellular deposits of amyloid-β, and neurofibrillary tangles [8]. The changes that occur are also associated with tau, a microtubule-associated protein that aids in microtubule assembly. The shape of tau proteins is altered in AD; thus, they are organized into neurofibrillary tangles. It was reported that the neuroinflammatory process (including the presence of amyloid-β plaques near glial cells and increased levels of inflammatory cytokines) is also involved in AD progression [9]. Decreased levels of the neurotransmitter acetylcholine are also observed in patients with AD [10]. Manganese acts as a chemical stressor in cholinergic neurons in a region-specific manner, causing breakdown of the cellular homeostatic mechanisms. In fact, a number of cholinergic synaptic mechanisms are putative targets for manganese activity: presynaptic choline uptake, quantal release of acetylcholine into the synaptic cleft, postsynaptic binding of acetylcholine to receptors, and its synaptic degradation by acetylcholinesterase. Moreover, manganese significantly influences astrocytic choline transport systems and astrocytic-acetylcholine-binding proteins [11].

#### 6.1.1. Micronutrients—Roles, Levels, Administration: Current Knowledge

Oxidative stress is one of the main players in the pathogenesis of neurodegeneration and impaired cognitive function [6]. The role of abnormal homeostasis of several micronutrients in AD progression is also known. Especially significant are lower plasmatic levels of vitamins A, B, C, D, and E, and Se, Cu, Zn, and increased levels of homocysteine (Hcy) [12,13,14,15]. 

Elevated levels of Hcy are associated with cognitive impairment. Because of the involvement of vitamins B9, B12, and B6 in Hcy metabolism, hypovitaminosis of these vitamins lead to hyperhomocysteinemia. Substitution of these vitamins helps to decrease the levels of Hcy. However, it has been reported that high doses of vitamins B9, B12, and B6 do not influence the cognition of patients with mild AD [6,16].

Other vitamins are also associated with AD pathogenesis: thiamine (B1) deficiency has been observed in patients with cognitive impairment, and supplementation improved the symptoms [17,18]. Vitamin B12 has a direct effect on tau proteins—it inhibits their fibrilization. Vitamin B3 (niacin) may have protective properties against AD and other types of cognitive decline. Prolonged treatment with vitamin E is in consideration as beneficial in the management of AD, but the outcome is still unclear. Vitamin A inhibits amyloid-β plaque formation. Vitamin D hypovitaminosis is acknowledged as a risk factor for AD. The pathogenic and therapeutic effects of vitamin D are not yet fully known, but its neuroprotective and anti-inflammatory functions are crucial. It was proposed as a potential therapeutic option for individuals with AD [19,20].

Detection of copper can be useful when diagnosing and preventing AD. 

Significantly higher levels of copper have been found in the brain tissue of AD patients. Copper supports oxidative stress, and it induces neurofibrillary tangle formation by tau hyperphosphorylation [21,22].

Another element involved in the pathogenesis of AD is zinc. Low plasmatic levels of zinc are repeatedly associated with decreases in learning ability and memory. Zinc status affects the progression of AD [23].

The neurotoxicity of manganese can also be associated with AD: it influences the function of astrocytes and the synthesis and degradation of glutamate. Monitoring of manganese can be one of the strategies for preventing AD. The correlation of higher selenium levels with higher cognitive abilities in the elderly was reported [24,25]. It is also believed that selenium deficiency may be linked to AD causation. Imbalance in iron metabolism and its accumulation also participates in AD development [26,27].

One of the proposed novel strategies for Alzheimer’s disease prevention is a diet rich in antioxidants [6].

#### 6.1.2. What to Do?

Here, the recommendation is the detection of available micronutrients, adjustment of deficiencies, and continued substitution in recommended doses. Pharmaceutical doses are recommended for faster adjustment of the deficiency, rather than for achieving a supranormal level. Homocysteine levels should be clarified in risk groups.

### 6.2. Parkinson’s Disease and Other Parkinsonian Syndromes

The clinical symptoms of Parkinson’s disease (PD) are caused by the degeneration of the dopamine-producing neurons in the basal ganglia, as well as nondopaminergic neurons (which cause motor and nonmotor symptoms). PD is associated with abnormal protein alpha-synuclein (αS) formations, manifested as Lewy bodies and Lewy neurites [28,29,30]. Similar to other neurodegenerative diseases, oxidative stress and neuroinflammatory processes are involved in the pathophysiology of PD [31].

#### 6.2.1. Micronutrients—Roles, Levels, Administration: Current Knowledge

Oxidative stress and lasting decrease in the levels of vitamins A, D, E, B1 and C play a key role in PD etiopathogenesis. For example, a decrease in thiamine causes a faster degeneration of dopaminergic neurons, and low levels of vitamin C impair levodopa absorption [32,33]. The administration of vitamin C improves the decreased absorption, and it reduces the toxicity induced by levodopa. Vitamin C can improve the absorption of levodopa in elderly patients with PD who show poor levodopa bioavailability [34]. Vitamin C prevents levodopa from breaking down in the periphery and increases its bioavailability and absorption in the brain. 

Previous studies have shown that plasma vitamin C levels in patients with PD were lower than those in healthy controls [35] and that patients with severe PD had lower lymphocyte vitamin C levels [36]. Vitamin C may be effective in patients with PD for purposes other than the stabilization of levodopa and carbidopa. Low levels of vitamin B6 are associated with a high risk for PD. Patients with Parkinson’s disease with olfactory impairment approximately 2 years before the symptoms had a diet with a low content of B1 and folate (although unproven but expected deficit) [37,38]. B1 and folate are responsible for the development and regulatory mechanisms of the olfactory system.

Chronical doses of vitamin E work in the same way, but they also improve the function of the dopaminergic receptors [39]. Vitamin D has anti-inflammatory effects and lowers oxidative stress [40]. Vitamin D deficiency is associated with dopaminergic neuronal death [41]. It was proven that adequate vitamin D serum levels might avoid Parkinson’s disease onset and possibly improve clinical outcomes. It was also reported that the serum concentration of vitamin D affects motor symptom severity in PD; additionally, sunlight exposure (>15 min/week) decreases the risk of PD [42].

Studies show that the effect of vitamin E is unrelated to age or sex, and higher levels of vitamin E decrease PD occurrence [43]. It is recommended to eat food rich in vitamin E because it reduces the risk of PD more than carotenoids or vitamin C. It was concluded that vitamin E supplementation protects against PD.

There is also a strong correlation between homocysteine levels and PD pathogenesis (homocysteine induces dopaminergic neuron death in PD patients) [44].

#### 6.2.2. What to Do?

Monitoring of folate levels (normally available) and B1 levels (more difficult to determine) can be indicated for the prevention of PD and the detection of the risk of PD development.

To lower the increased Hcy levels induced by conventional PD medication with levodopa, it is recommended to give the patient folic acid. The level of Hcy should be monitored and regulated, which can prevent the progression of Parkinson’s disease.

### 6.3. Amyotrophic Lateral Sclerosis and Other Forms of Motor Neuron Disease

The spectrum of the clinical symptoms of amyotrophic lateral sclerosis (ALS) includes upper (motor cortex—weakness, hyperreflexia, spasticity) and lower (motor neurons in the brainstem and spinal cord—weakness, atrophy, or amyotrophy; fasciculations) symptoms, which are induced by the degeneration of motor neurons [45]. Intracellular inclusions, such as Bunina bodies or TDP-43 accumulation, can be found in nerve cells. Several theories concerning the etiology of motor neuron disease have been proposed: abnormal RNA processing, disorders of protein quality control, excitotoxicity, cytoskeletal derangements, mitochondrial dysfunction, viral infections, apoptosis, growth factor abnormalities, and inflammatory responses [46,47]. 

#### 6.3.1. Micronutrients—Roles, Levels, Administration: Current Knowledge

In patients with ALS, it is beneficial to administer vitamins D and E. Vitamin D deficiency is an independent prognostic factor for ALS development.

There are insufficient data to report the efficacy of vitamin D supplementation in ALS patients. 

#### 6.3.2. What to Do?

It is beneficial to monitor the serum levels of 25-OH vitamin D and vitamin E to discover deficiencies [48]. In hypovitaminosis, it is recommended to substitute vitamins as adjuvant therapy.

### 6.4. Wilson’s Disease

Wilson’s disease (WD) is an autosomal-recessive metabolic disease based on copper accumulation in the liver, brain, and other organs because of mutations in the ATP7B gene [49]. WD can manifest as movement impairments (Parkinsonian syndromes, dyskinesias), behavioral disorders, intellectual disorders, and hepatic symptoms [50].

A proven treatment strategy in WD with lower copper blood levels is the oral administration of zinc. The result of hyperzincemia is the production of metallothionein, which helps to decrease free zinc concentrations; however, copper has the highest affinity for metallothionein. High levels of zinc reduce the levels of copper.

#### 6.4.1. Micronutrients—Roles, Levels, Administration: Current Knowledge

Zinc can be administered as sulfate, acetate, or gluconate salts.

#### 6.4.2. What to Do?

To assess WD early on, it is recommended to monitor the plasmatic levels of Cu, Zn, and ceruloplasmin. 

Due to the increased consumption of vitamin B6, it is necessary to permanently administer a small dose of pyridoxine (40–60 mg/day) to patients. Zinc is the most common alternative to penicillamine treatment. The basic mechanism of action is the stimulation of metallothionein synthesis in the intestinal cells and the creation of a mucosal block for copper resorption, thereby increasing the excretion of copper in the stool.

Zinc treatment is associated with few side effects. The most common side effect of zinc treatment is gastric irritation, which sometimes may be helped by concomitant intake of protein or use of proton pump inhibitors [51]. Occasionally, the irritation can be severe, resulting in erosions and ulcers. Other side effects that rarely occur include biochemical pancreatitis, resulting in elevated pancreatic enzymes (lipase/amylase) without clinical symptoms. With long-term use, the issues of overtreatment and zinc-induced copper deficiency can occur. This typically presents with anemia, neutropenia, as well as neurologic symptoms, including sensory or motor sensory neuropathies and myelopathies, which normally reverse with the correction of the deficiency.

Treatment monitoring while on zinc therapy should include 24 h urine copper and zinc excretion and serum ALT level. 

### 6.5. Huntington’s Disease

Huntington’s disease (HD) is an autosomal-dominant neurodegenerative disorder manifested as defects in voluntary motor control, involuntary movements, behavioral disorders, and cognitive impairment [52]. HD is caused by a mutation in the huntingtin gene, which results in neuronal dysfunction and death through several mechanisms [53].

#### 6.5.1. Micronutrients—Roles, Levels, Administration: Current Knowledge

The development of motoric abnormality and dementia in Huntington’s disease is attributed to neurodegenerative changes caused by oxidative stress and neuroinflammation. Administration of vitamin C and vitamin D during adjuvant therapy should slow down the progress of the disease and prevent postural instability. Coenzyme Q10 and vitamins A, B1, B3, and E are all valued for their significant neuroprotective activity. Individuals with daily supplementation of vitamin C, β-carotene, vitamin E, selenium, and zinc had better episodic memory scores six years after.

It was shown that people with HD have a high caloric intake, so it is necessary to increase micronutrient supplementation (especially vitamin B).

HD patients consuming the Mediterranean diet, which contains greater amounts of micronutrients (vitamins C, B1, B3, B6, and E, selenium, and zinc) show improvement in cognitive and motor abilities, and their quality of life is reportedly better [54].

#### 6.5.2. What to Do?

It is beneficial to supplement antioxidant micronutrients as adjuvant therapy.

### 6.6. Myasthenia Gravis

Myasthenia gravis (MG) is an autoimmune disorder affecting the neuromuscular junction, which manifests as skeletal muscle weakness. The molecular basis of the disease consists of the immunoglobulin G class of antibodies being directed against certain molecules of the postsynaptic neuromuscular junction (in most cases, nicotinic acetylcholine receptors), which causes impaired neuromuscular transmission [55]. Individuals with MG often suffer from other autoimmune diseases (thyroid disease, systemic lupus erythematosus, rheumatoid arthritis).

#### 6.6.1. Micronutrients—Roles, Levels, Administration: Current Knowledge

Significantly lower plasmatic concentrations of 25-OH vitamin D were observed in patients with MG [56]. Moreover, lower levels in patients with myasthenic crisis were reported compared with those of patients with MG without myasthenic crisis. This means that vitamin D is an important factor in MG progression, and supplementation of vitamin D can be beneficial in myasthenic crisis prevention.

There was no significant difference when comparing vitamin B12 and folic acid levels between myasthenic crisis and noncrisis periods [1].

Because MG management requires prolonged pharmacotherapy with corticosteroids and other immunosuppressants, there are various adverse effects (osteoporosis, infections). Furthermore, female patients suffer often from anemia [57]. 

#### 6.6.2. What to Do?

It should be advised to routinely monitor the plasmatic level of 25-OH vitamin D and to correct the level when necessary. In female patients, hemoglobin and Fe levels should be monitored for the best outcome.

### 6.7. Multiple Sclerosis and Other Demyelinating Diseases

Multiple sclerosis (MS), neuromyelitis optica spectrum disorders (NMOSDs), and myelin oligodendrocyte glycoprotein antibody disease (MOGAD) are disorders affecting optic nerves, brain, and spinal cord and lead to various clinical symptoms [58]. The main features of these disorders are immune-mediated demyelination and degeneration of neurons. In MS, there are inflammatory-induced plaques consisting of CD8+ and CD4+ T cells [59]. New therapeutic strategies include immunomodulation, preservation of the quantity and integrity of neurons, reduction in oxidative stress, and support of the hematoencephalic barrier [58,60].

#### 6.7.1. Micronutrients—Roles, Levels, Administration: Current Knowledge

It has been reported that the incidence of MS depends on the number of hours spent in the sunshine [61,62]. The levels of 25-OH vitamin D may be linked to disease activity. Furthermore, adequate plasmatic levels of 25-OH vitamin D are associated with lower risks of MS and relapse. The effects of the supplementation of vitamin D on MS relapse are not yet clear [63,64]. 

Increased levels of Hcy and low levels of vitamin E were observed in patients with MS [65]. The administration of vitamins A and E may be beneficial for their anti-inflammatory and antioxidative effects; vitamin A also helps with remyelination.

#### 6.7.2. What to Do?

It is important to monitor the plasmatic levels of 25-OH vitamin D. Its deficiency should be prevented, and a level of 100 nmol/L (50 ng/mL) should be maintained.

There is no clear guideline for the best dosage, but vitamin D3 supplementation with dosages from 1000 to 2000 IU/day to 10,400 IU/day are considered well tolerated and may be recommended. Dosage should be determined according to the patient’s residence (higher doses for north Europe and North America lower doses for Australia and south Europe).

### 6.8. Epilepsy

Neurodegeneration promoted by oxidative stress is one of the possible mechanisms of epileptogenesis. Elevated lipid peroxidation was also observed in epileptic patients. The development of epilepsy was associated with mitochondrial dysfunction [66]. 

#### 6.8.1. Micronutrients—Roles, Levels, Administration: Current Knowledge

Vitamin E is an antioxidant, anti-inflammatory, and neuroprotective substance, and it has been shown that daily supplementation of 400 IU reduced the frequency of seizures by approximately 60% compared with a placebo. Thus, chronic administration of vitamin E can be beneficial when treating recurrent epilepsy. 

Vitamin D also appears to be a promising candidate for adjuvant therapy for epilepsy. 

There are cases of epilepsy dependent on vitamin B6 deficiency, which is then treated with high doses of B6 [67].

It was also reported that treatment combining pyruvate, vitamin C, and vitamin E could reduce the frequency of seizures [68].

#### 6.8.2. What to Do?

Monitoring of plasmatic levels of 25-OH vitamin D is recommended. Patients with epilepsy benefit from the supplementation of antioxidant micronutrients as adjuvant therapy.

### 6.9. Ischemic Stroke

Ischemic stroke is caused by thrombosis, embolization, or systemic hypoperfusion. Hypoperfusion then leads to severe stress; disruption of the plasma membrane; leaking of cellular contents; and, eventually, to loss of neuronal function and cell death. Furthermore, inflammation, acidosis, free-radical-mediated toxicity, cytokine-mediated toxicity, complement activation, impairment of the hematoencephalic barrier, oxidative stress, and other factors contribute to hypoxic damage [69,70].

#### 6.9.1. Micronutrients—Roles, Levels, Administration: Current Knowledge

Several micronutrients have a significant impact on decreasing the occurrence of stroke: vitamins B6, C, and D; β-carotene; folic acid; magnesium; and potassium [71,72]. When evaluated separately, vitamins C and E, magnesium, and potassium protected against ischemic stroke, but made no difference against hemorrhagic stroke.

Vitamin C protects the cardiovascular system, and it reduces the risk of stroke due to its anti-inflammatory, antioxidant, and endothelial protective effects [73,74].

Again, elevated levels of Hcy increase the risk of stroke.

#### 6.9.2. What to Do?

Plasmatic levels of Hcy and folate should be monitored in the at-risk population [75]. Adjuvant therapy with micronutrient supplementation is recommended [76,77].

It remains under investigation whether vitamin C supplementation to decrease oxidative stress in patients with stroke is beneficial, and what amount and source of vitamin C are most adequate [78].

### 6.10. Myopathy

Myopathy is a muscle disorder without a connection to innervation or neuromuscular junction defects. The etiology of myopathy can be varied: it includes congenital, idiopathic, infectious, metabolic, inflammatory, endocrine, or drug-induced myopathy. Idiopathic myopathies can be associated with autoimmune diseases like systemic lupus erythematosus, rheumatoid arthritis, dermatomyositis, etc. [79]. In some cases, inflammatory myopathy can be linked to statin usage [80]. Alcohol should be considered as a cause of acute myopathies [81].

#### 6.10.1. Micronutrients—Roles, Levels, Administration: Current Knowledge

Selenium deficiency manifests as a cardiac and skeletal myopathy.

Vitamin E deficiency is associated with coordination disorders, neuropathy, or muscle weakness. These symptoms improve after a few months of vitamin E supplementation (200 mg/day) [82].

In cases of mitochondrial myopathy, it is important to improve mitochondrial metabolism using vitamins B, C, and E; selenium; zinc; and coenzyme Q10.

#### 6.10.2. What to Do?

Management of potential myopathy can include the monitoring of myoglobin, creatinine, and creatine kinase to assess the severity of muscle damage and to detect rhabdomyolysis. 

Adjuvant multivitamin therapy for oxidative stress reduction can be recommended.

### 6.11. Neuropathy

A frequent side effect of neurotoxic chemotherapy (taxanes, vinca alkaloids, platinum compounds, etc.) is chemotherapy-induced neuropathy [83,84].

More than 50% of patients with long-standing diabetes suffer from diabetic neuropathy; its most common manifestation is diabetic sensory neuropathy [85].

Chronic alcohol abuse can also lead to neuropathy caused, e.g., by direct alcohol toxicity, nutritional deficiencies (vitamins B1 and B12), and hepatic cirrhosis. It affects small and large fibers, and its presentation widely varies [86].

#### 6.11.1. Micronutrients—Roles, Levels, Administration: Current Knowledge

Micronutrients crucial for the right function of the peripheral nervous system are the B vitamins (especially B1, B6, B9, and B12), E, and copper.

Nutritional neuropathies can present as acute, subacute, or chronic and as demyelinating or axonal. There can also be a combination of peripheral neuropathy with myelopathy—myeloneuropathy—which is mostly caused by vitamin B12 or copper deficiency [83].

The clinical manifestation of thiamine (B1) deficiency involves the heart as well as the central and peripheral nervous systems. The symptoms may be presented as Wernicke encephalopathy—a characteristic triad of ataxia, ophthalmoparesis, and encephalopathy [87].

Pyridoxine (B6) deficiency may involve loss of sensation or paresthesia. On the other hand, pyridoxine excess may lead to neuropathy.

Cobalamin (B12) plays a part in the myelination of the central and peripheral nervous systems; its deficiency leads to demyelination of the dorsal and lateral columns as well as of the peripheral and optic nerves. A plasmatic concentration of less than 150 pg/mL is recognized as a deficiency. Folate deficiency can lead to myelopathy or optic neuropathy.

Vitamin E deficiency most commonly affects the central nervous system but rarely can be manifested also as peripheral nerve disorder.

Copper is involved in cellular transporters, mitochondrial oxidative metabolism, biosynthesis of neurotransmitters, and maintaining the function of the nervous system.

In alcohol-related neuropathy (ALN), the toxic effects of alcohol are altered by thiamine deficiency. ALN is painful sensory neuropathy; thiamine deficiency mainly affects large fibers and is associated with decreased mobility.

B vitamins are particularly important in the management of diabetic polyneuropathy [88]. B1 helps with nerve regeneration and protects nerves against oxidative stress, which leads to normal pain sensations. B6 contributes to neurotransmitter synthesis and sensor nerve function restoration. B12 promotes nerve survival, remyelination, and maintenance of myelin sheaths, which results in improvement or even complete recovery of nerve function.

Vitamin D deficiency is highly associated with diabetic peripheral neuropathy.

#### 6.11.2. What to Do?

Supplementation of micronutrients with aberrant homeostasis is recognized as beneficial. For instance, treatment with 40,000 IU/week of vitamin D significantly improved symptoms as well as cutaneous microcirculation [89]. It also led to a decrease in IL-6 and an increase in IL-10 serum levels. Generally, a dose of 800 to 2000 IU/day of vitamin D is recommended for all diabetic patients. 25-OH vitamin D serum levels of 75 nmol/L have to be maintained because mineralization defects are prevented at these levels.

### 6.12. Restless Leg Syndrome

Restless legs syndrome (RLS) is a movement disorder manifesting as an uncontrollable urge to move as a result of painful sensations in the legs with a release of movement. The pathophysiology is not clear, but genetic components, neurotransmitter dysfunction and aberrant brain iron homeostasis are all contributors [90,91].

#### 6.12.1. Micronutrients—Roles, Levels, Administration: Current Knowledge

It was reported that iron deficiency may cause or worsen RLS; approximately 25% of pregnant women experience RLS during pregnancy, but it is often resolved after birth.

#### 6.12.2. What to Do?

It is important to indicate blood tests to eliminate other reasons for RLS development such as kidney failure, low iron levels [92], or pregnancy.

### 6.13. Injury of the Central Nervous System

Intraparenchymal hemorrhagic stroke and brain and spinal cord contusions are associated with several pathways through which nerve tissue damage is worsened. In addition to the initial mechanical injury, secondary mechanisms play an important role, such as water transport dysregulation within the central nervous system (CNS) (cytotoxic edema) and the disruption of blood–spinal cord/brain barrier (vasogenic edema), loss of ATP, oxygen deprivation, excitotoxicity, and ionic imbalance. Furthermore, reperfusion to the damaged site introduces more oxygen free radicals and immune cells, which worsens oxidative stress. These factors contribute to exacerbation and remodeling of the injury site, which inhibits neurological recovery [93,94,95,96].

#### 6.13.1. Micronutrients—Roles, Levels, Administration: Current Knowledge

After the insult, secondary mechanisms like oxidative stress, inflammation, and autophagy disruption contribute to decreasing the potential for neurological recovery [94].

Vitamin B3 has a key role in the development of neurons and their survival [97,98]. The effects are still present even if administered up to 24 h after the insult. It was reported that it is beneficial to administer vitamin B3 with other compounds (such as progesterone) to increase recovery, reduce tissue loss, as well as modulate inflammatory and immune responses.

#### 6.13.2. What to Do?

It is recommended to administer adjuvant antioxidant multivitamin treatment. When giving vitamin B3, it should be combined with other natural compounds.

### 6.14. Peripheral Nervous System Injury

Traumatic insult to a peripheral nerve may result in losses of structure and sensory and motor function [99,100]. The outcome is usually poor because the injury leads to the rupture of the axon, myelin sheath, and connective tissue [101,102].

#### 6.14.1. Micronutrients—Roles, Levels, Administration: Current Knowledge

Neuronal viability is maintained by neurotropic B vitamins (B1, B6, B12). B1 is an antioxidant at the site of the insult, B6 improves nerve metabolism, and B12 maintains myelin sheaths [103]. A deficiency of these vitamins may lead to permanent nerve degeneration and pain, resulting in peripheral neuropathy [104,105].

#### 6.14.2. What to Do?

It was reported that the supplementation of vitamins B1, B6, and B12 is beneficial.

**Table 2 nutrients-15-04129-t002:** Monitoring and supplementation of micronutrients recommended in various neurological disorders.

Disease/Disorder	Supplementation with Proven Benefit; Pharmacological Dose, If Applicable *	Adjuvant Therapy	Deficiency Detection, Monitoring
Alzheimer’s disease (AD)	Vitamins A, D, E, C, B; selenium; zinc. FD vitamin B1 (cognitive impairment with dementia)	DRA	Hcy, 25-OH vitamin D, Cu, Zn
Parkinson’s disease (PD)	Vitamins A, D, E, C, B1, B6; folic acid	DRA	B6, 25-OH vitamin D, vitamin E, Hcy
Amyotrophic lateral sclerosis (ALS)	Vitamins E, D, B, C	DRA	25-OH vitamin D, vitamin E
Wilson’s disease (WD)	FD Zn	DRA	Cu, Zn, ceruloplasmin
Huntington’s disease (HD)	Coenzyme Q10; vitamins A, D, E, C, B1, B3; biotin; Se; Zn (pyruvate)	DRA, MD	25-OH vitamin D
Myasthenia gravis (MG)	Vitamin D	DRA	AChRs, MuSK, 25-OH vitamin D, Fe (susp. anemia)
Multiple sclerosis (MS)	Vitamins A, D, E,	DRA	25-OH vitamin D, vitamin E, Hcy
Epilepsy (Epi)	Vitamins E, D, B6, C; omega-3 PUFA; pyruvate	DRA	25-OH vitamin D
Ischemic stroke (IS)	Vitamins A, D, E, B6; folic acid; vitamin C; Mg; K	DRA, MD	Hcy
Myopathy (MP)	Vitamins E, C; Se; Zn; coenzyme Q10	DRA	Se, creatinine, myoglobin
Neuropathy (NP)	Vitamins D, E, B1, B6, B12; folic acid; Cu	DRA	B12, Cu, B1
Restless leg syndrome (RLS)	Fe (in case of deficit)	DRA	Fe
Injury central nervous system (ICNS)	Vitamin B3 (niacin)	DRA	
Injury peripheral nervous system (IPNS)	Vitamins B1, B6, B12, D, E	DRA	Vitamin E, vitamin B12, 25-OH vitamin D
Sarcopenia (SA)	B vitamins, vitamin D	DRA	Vitamin B1, 25-OH vitamin D

DRA: diet rich in antioxidants, MD: Mediterranean diet, Hcy: Homocysteine, AChrs: acetylcholine receptors, MuSK: muscle-specific kinase. * Guideline daily amounts.

## 7. Summary

Oxidative stress, neuroinflammation, aging, an increase in homocysteine levels, and a permanent decrease in the plasma levels of some micronutrients are the main common factors influencing the emergence and progression of neurodegenerative diseases. Low levels of micronutrients reduce the activity of antioxidant enzymes, which may lead to DNA, protein, and fatty acid oxidation and crosslinking, along with mitochondrial ATP depletion, therefore contributing to neurodegenerative disorders. Micronutrient deficiencies have been linked to changes in bacterial species in the human gut microbiota, affecting the host regulation of immune responses. The activity of the gut microbiota significantly contributes to the host’s immune health and is linked to the development of many diseases. Vitamin D supplementation has been shown to increase gut microbial diversity significantly. Given the shared deficiency of some micronutrients and oxidative stress in all neurodegenerative diseases, the question of the amount and frequency of the dose is similarly shared among micronutrients. It has been repeatedly reported (sometimes with controversial results) that the administration of certain micronutrients may be beneficial. But because of the large diversity of micronutrients, it is very difficult to establish the amount and frequency of the dose appropriate for individual micronutrients. There is no other way but to follow universal guidelines and wait for the outcome of further research in this specific field. Multivitamin supplementation shows solid therapeutic potential for neurodegenerative diseases compared with single-vitamin-based options. Multiple signaling pathways capable of boosting the antioxidative response are involved in the multivitamin approach. It is also possible that such adjuvant agents are most effective for prevention rather than treatment. 

## 8. Highlights

Micronutrients have a crucial and irreplaceable role in intermediary metabolism. They are an integral part of the therapy of neurological disorders for their neuroprotective, antioxidant, and anti-inflammatory effects.

The frequently difficult-to-establish clinical effect of micronutrient administration does not mean that the administration is dispensable. Clear pathophysiologic consequences are proof of the dangerousness of deficiency.

The mechanism through which the systemic inflammatory response of the organism to the infectious or noninfectious noxa is influenced is not completely clear for most of the micronutrients. The relationship between micronutrient depletion and the reduction in microbiome diversity has been established.

Early diagnosis of dysregulations and adequate substitute micronutrient therapy are necessary for the reduction in oxidative stress, improvement in energetic substrate metabolization, and the enabling of proper enzymatic activities. Therefore, it is a crucial therapy in the prevention of CNS damage during development, nutritional neuropathies, and neurodegenerative diseases.

Pharmacological micronutrient superdoses are effective only for B1 administration during a deficit. 

## Figures and Tables

**Table 1 nutrients-15-04129-t001:** DDD of vitamins and trace elements for enteral and parenteral administration (ESPEN 2022). All values are per day.

	PN Home and Long-Term A	PN High Requirements ^a^ B	EN in 1500 kcal ^b^ C	EN High Requirements in 1500 kcal ^c^	DRI per Day Age 31–70 Years	Min–Max per 1500 kcal ^d^
**Trace elements**						
Chromium	10–15 μg	15 μg	35–150 μg	200 μg	20–35 μg	18.75–225 μg
Copper	0.3–0.5 mg	0.5–1.0 mg	1–3 mg	Same as C	0.9 mg	0.9–7.5 mg
Fluoride	0–1 mg	Same as A	0–3 mg	3–4 mg	3–5 mg (AI)	0–3 mg
Iodine	130 μg	Same as A	150–300 μg	Same as C	150 μg	97.5–525 μg
Iron	1 mg	Same as A	18–30 mg	30 mg	8 mg (18 mg F 19–50 years)	7.5–30 mg
Manganese	55 μg	Same as A	2–3 mg	Same as C	1.8–2.3 mg	0.75–7.5 mg
Molybdenum	19–25 μg	Same as A	50–250 μg	250 μg	45 μg	52.5–270 μg
Selenium	60–100 μg	150–200 μg	50–150 μg	200 μg	55 μg	37.5–150 μg
Zinc	3–5 mg	6–12 mg	10–20 mg	20 mg	8–11 mg	7.5–22.5 mg
**Lipo-soluble vitamins**						
A Retinol ^e^	800–1100 μg	1100 μg	900–1500 μg	1500 μg	700–900 μg	525–2700 μg
D3 Cholecalciferol	200 IU/5 μg	800–1000 IU/20–25 μg	25 μg	30 μg	15–20 μg	7.5–37.5 μg
E α-tocopherol	≥9 mg	20 mg	15 μg	40 mg	15 mg	7.5–45 mg
K1 Phylloquinone	150 μg	1–10 mg ^f^	120 μg	Same as C	90–120 μg	52.5–300 μg
**Water-soluble vitamins**	Provide at least ^g^:		Provide at least ^g^:			
B1 Thiamine	2.5 mg	100–200 mg	1.5 mg	100 mg	1.1–1.2 mg	0.9–7.5 mg
B2 Riboflavin	3.6 mg	10 mg	1.2 mg	10 mg	1.1–1.3 mg	1.2–7.5 mg
B3 Niacin	40 mg	Same as A	18 mg	40 mg	11–16 mg	13.5–45 mg
B5 Pantothenic acid	15 mg	Same as A	5 mg	7.5 mg	5 mg	2.25–22.5 mg
B6 Pyridoxine	4 mg	6 mg	1.5 mg	7.5 mg	1.5–1.7 mg	1.2–7.5 mg
B7 Biotin	60 μg	Same as A	30 μg	75 μg	30 μg (AI)	11.25–112.5 μg
B9 Folic acid	400 μg	600–1000 μg	330–400 μg DFE	500 μg	400 μg DFE	150–750 μg
B12 Cyancobalamin	5 μg	Same as A	>2.5 μg	7.5 μg	2.4 μg	1.05–10.5 μg
C Ascorbic acid	100–200 mg	200–500 mg	100 mg	200 mg	75–90 mg	33.75–330 mg

*Abbreviations*: EN = enteral nutrition, PN = parenteral nutrition, AI = adequate intake, DFE = dietary folate equivalent. Note 1: Major burns and some gastrointestinal conditions (fistulae) may have losses that are not covered by the above increased doses. Note 2: Cobalt is provided as vitamin B12. ^a^ Increased requirements may occur in patients with increased losses such as gastrointestinal losses, continuous renal replacement therapy, patients depleted before commencing PN, and in pregnancy. ^b^ In case of higher nutrient delivery (2000 kcal per day or more), exceeding this recommendation is not exposing the patient to any risk considering upper tolerable levels. ^c^ Increased requirements during critical illness and in patients with acute admission with malnutrition (NRS ≥5); intended for max 15 days as repletion, to avoid requiring i.v. supply. ^d^ This column indicates the minimal and maximal trace element contents for 1500 kcal/day. ^e^ Retinol includes retinol and retinyl ester. ^f^ High dose administered in case of coagulopathy. ^g^ For water-soluble vitamins, amounts recommended are minimum amounts, and more can usually be safely delivered.

## Data Availability

Not applicable.

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
