# Peer review of "The Role of Micronutrients in Neurological Disorders"

_nutrients, 2023, doi:10.3390/nu15194129_

Round 1

Reviewer 1 Report

In the present review, Lahoda Brodska et al. have considered the role of micronutrients like vitamins and trace elements in defined neurological disorders (Alzheimer´s disease, Parkinson´s disease, ALS, Myastenia gravis, Multiple Sclerosis, etc.). The topic of the review is very interesting and up-to-date. The review is clearly structured and the included table 2 is very informative.

However, I have some comments that I would like to address:

-        The abstract and Keywords are missing.

-        The table 1 should be included.

-        The review can be structured even better by numbering the headings to the information of individual diseases (1. Alzheimer's disease, 2. Parkinson's disease, etc.).

-        Line 205: only “Vitamine D hypovitaminosis” should be marked bold.

-        According to table 2: a paragraph on Sarcopenia is missing so far.

-        The last sentence of the last paragraph (line 584-585) should removed from the “Highlights”.

Author Response

However, I have some comments that I would like to address:

-        The abstract and Keywords are missing. - Corrected

-        The table 1 should be included. Added, again in better resolution, we do not have table in word, only in PDF, please see source article

-        The review can be structured even better by numbering the headings to the information of individual diseases (1. Alzheimer's disease, 2. Parkinson's disease, etc.). Correted

-        Line 205: only “Vitamine D hypovitaminosis” should be marked bold. Correted

-        According to table 2: a paragraph on Sarcopenia is missing so far.  Please,delete sarcopenia in the table

-        The last sentence of the last paragraph (line 584-585) should removed from the “Highlights”. Correted

Reviewer 2 Report

The manuscript is an interesting and important set of data, as well as practical recommendations for the prevention of neurological disorders with the help of micronutrients. In addition, data are given on what neurological disorders can occur with a lack of certain micronutrients.

However, there are a number of notes:

1. Affiliations are issued without specifying the address.

2. There is no abstract in the article.

3. There are only 3 references to literary sources in the introduction. A number of statements are not supported by references. The introduction needs to be improved. For some reason the links are in parentheses. Must be square.

4. The introduction is the rationale for the purpose of the study, ends with its statement. Target is missing. Authors must indicate the purpose of the study.

5. Tables are inserted into the text as a picture... It is necessary to arrange them according to the requirements of the journal. Line 158 - references behind a dot.

Author Response

1. Affiliations are issued without specifying the address. Corretetd
2. There is no abstract in the article. Corretetd
3. There are only 3 references to literary sources in the introduction. A number of statements are not supported by references. The introduction needs to be improved. For some reason the links are in parentheses. Must be square. Referencing 3 comprehensive papers is quite sufficient, the other papers just repeat the same information. The use of parentheses is consistent throughout the text. If you think it should be different, please modify it according to the conventions of your journal. Thank you for your understanding and cooperation.
4. The introduction is the rationale for the purpose of the study, ends with its statement. Target is missing. Authors must indicate the purpose of the study. Corrected
5. Tables are inserted into the text as a picture... It is necessary to arrange them according to the requirements of the journal. Line 158 - references behind a dot. The table can be taken from the source publication we attach (ESPEN micronutrients guidelines). Table 2 is in word format.